# Fostering the Ecosystem of AI for Social Impact Requires Expanding and Strengthening Evaluation Standards

**Bryan Wilder**
Machine Learning Department
Carnegie Mellon University

**Angela Zhou**
Department of Data Sciences and Operations
Department of Computer Science
University of Southern California

## Abstract

There has been increasing research interest in AI/ML for social impact, and correspondingly more publication venues have refined review criteria for practice-driven AI/ML research. However, these review guidelines tend to most concretely recognize projects that simultaneously achieve deployment and novel ML methodological innovation. We argue that this introduces incentives for researchers that undermine the sustainability of a broader research ecosystem of social impact, which benefits from projects that make contributions on single front (applied *or* methodological) that may better meet project partner needs. Our position is that researchers and reviewers in machine learning for social impact must simultaneously adopt: 1) a *more expansive conception of social impacts beyond deployment* and 2) *more rigorous evaluations of the impact of deployed systems*.

## 1 Introduction / Position

There has recently been increasing interest in research on machine learning or artificial intelligence for social impact (AISI). AISI differs from the field as a whole through an emphasis on identifying and solving applied problems, with positive impact on society as the primary goal. (We interchangeably use the shorthand AI or ML for social impact, or AISI throughout). Previous books and position papers highlight that impact often requires collaborations with researchers in other disciplines or with partner organizations (e.g. nonprofit or government actors), new problem formulations that serve their needs, and confronting substantial last-mile challenges to deploy and evaluate ML systems [42, 41, 37]. For example, [12] collaborated with the government of Greece to develop and deploy a novel reinforcement learning system for targeting COVID-19 testing of incoming travelers. Or, [39] collaborated with a food rescue nonprofit, developing a new machine learning + optimization system to improve their volunteer dispatching. There are many other such projects, including training pretrial risk assessment models in criminal justice [9] or deploying early sepsis detection systems in a hospital [14]. Previous work argues, and we agree, that such projects require a distinctive set of research practices not present in "traditional" machine learning research. Such additional processes are essential both to impact many critical application domains and to enrich the field of machine learning with new problem formulations that come from working closely with external partners and testing ideas in the real world.

This vision has been sufficiently compelling that research in machine learning/AI for social impact has grown significantly over the past decade, with dedicated tracks at major conferences such as AAAI and IJCAI [1, 6], courses [4, 2, 3], summer programs [5, 7], and more. However, we argue that the present academic ecosystem of the field suffers a poorly-defined vision of what constitutes "impact" and how impact can be rigorously evaluated. In particular, we argue that a (1) singular focus on deployment of novel ML systems as the pathway to impact has foreclosed other important

39th Conference on Neural Information Processing Systems (NeurIPS 2025) Position Paper Track.

mechanisms for academic work to have impact; and, (2) even within the predominant deployment frame, the field lacks rigor in evaluating the results of deployments.

**Accordingly, our position is that researchers and reviewers in machine learning for social impact must simultaneously adopt**: 1) a *more expansive conception of social impacts beyond deployment* and 2) *more rigorous evaluations of the impact of deployed systems*.

Our position responds to a common underlying conception of the "ideal project" as one that engages with the needs of practitioners, develops technically novel machine learning methods to meet those needs, and then deploys a system based on the new methods. This goal is pervasive across previous positions articulating visions for the field, reviewing guidelines, and educational materials. For example, the IJCAI Multi-Year Track On AI And Social Good evaluates papers based on "contribution to state-of-the-art AI" and "collaboration with stakeholders/partners", with "potential deployment/implementation opportunities in the future" as a key desiderata [6]. The AAAI Special Track on AI for Social Impact's review criteria [1] prioritize papers that are deployed or close to deployment (the "Scope and promise for social impact" criterion) and which are methodologically novel (the "Novelty of approach") criterion. This is not to say that existing venues only accept work that deploys novel ML methods – just that it is the single clearest way for work to demonstrate excellence and score high on all criteria. It is less clear how reviewers should assess projects that depart from this framing.

We do not dispute that the successful deployment of novel methods constitutes an appealing and valuable vision. It is an important ideal to strive for. However, exclusive focus on this ideal generates negative consequences for researchers, partner organizations, and the field as a whole. First, it neglects creative ways to make valuable scientific contributions even within the context of projects that do not achieve the entire ideal all at once – projects that deploy a ML method that is not novel, or develop practice-motivated methods without reaching deployment. Second, deployment is often envisioned as the "finish line" for successful projects. However, current practices in the field do not always demand or reward highly rigorous evaluations of whether deployments were truly impactful. Combined, these incentives push researchers towards fitting all projects into the same idealized framework, even if the contingencies of real-world projects and the needs of partner organizations would sometimes lead to other (also scientifically valuable) kinds of contributions.

We propose three steps to foster a healthier, more scientifically rigorous ecosystem of research in machine learning for social impact. We focus on structural critiques and encouraging systemic reforms, rather than drawing attention to shortcomings in any particular review, review process or project. First, the field should recognize contributions where researchers contribute to the use of machine learning in practice without developing technically novel methods. Second, the field should recognize contributions where novel methodologies have strongly plausible impacts on practice, even if they are not immediately deployed by the same team. Third, while deployments of novel methods should still be highly valued, the field should adopt higher standards of rigor in evaluating such deployments. We first elaborate on each of these steps, articulating specific kinds of contributions that papers outside the current "idealized project" can make, and suggesting best practices for the design and analysis of field trials (deployments) of new ML systems. Finally, we illustrate how adoption of these suggestions by authors and reviewers can improve incentives in the overall ecosystem for research in machine learning and social impact, driving both scientific progress and better results for partner organizations who place their trust in researchers.

## 2   Non-method contributions: Improving partners' usage of machine learning

Machine learning for *social* impact research often involves partnering with organizations focused on the application of interest–e.g. nonprofit organizations or government agencies delivering services. Distinctly from what is often considered "applied ML", these partner organizations often do not have a great deal of in-house expertise related to machine learning. Recent conferences have introduced a "societal impacts of ML" category, but this label does not clearly distinguish papers that study general formulations from practice-driven projects that originate in partner collaborations. A key benefit to the partner in collaborating with researchers may be to gain access to such expertise. Importantly, this benefit often does not depend on developing novel methods at all – researchers have a positive impact just through advising on the right way to use and evaluate existing, "standard" machine learning tools within the context of the application domain. This process is a critical but often-overlooked

part of the process of leveraging ML for social impact. Beyond the immediate benefits of a project, collaborations help build technical capacity within partner organizations. This is especially important for the maintainability and adaptability of the initial product, and the partner's ability to develop similar projects in the future. Finally, such projects are valuable opportunities for the researcher to learn about the application area and inform future problem formulation.

However, this kind of impact is invisible within current frameworks for what counts as an academic contribution. Machine learning publication venues are often reluctant to accept papers that purely focus on an application of machine learning, without novel methods. This reluctance is in part because of a legitimate question – what scientific importance does such work have for the field of machine learning, and are machine learning venues the appropriate place to publish? Here, we describe several ways that researchers can articulate a scientific contribution as a result of work with a partner organization that informs their use of machine learning. We argue that the field of machine learning should embrace such contributions, while also delimiting some kinds of contributions that are best reviewed and published within application-domain journals.

Broadly, we argue that, researchers can make scientific contributions to machine learning as a discipline by studying how machine learning is used by the partner and the impact of its introduction. Examples of potential contributions to the field of machine learning include shedding light on any of the following relatively under-explored questions:

- How does ML enter and inform organizational decision processes, and how do modeling choices alter how it is used?

- What is the value of machine learning in a particular domain: did improving prediction lead to improved outcomes or other value for the partner? If so, how?

- What constraints and objectives shape how it is possible to use machine learning in a particular setting? Are there particular problem formulations or classes of methods that are not well-suited, despite seeming relevant? Currently authors tend to explain the problem formulation they settled on, but other researchers can also learn a great deal from discussion of alternatives that were considered.

- What is the right level of complexity in problem formulation? Researchers often decide how many complicating factors to include in the formal problem that an algorithmic system solves. However, it is often not clear when added complexity is necessary, or when a simpler version of the problem formulation would have sufficed.

Empirical evidence across domains is needed to inform problem formulation; it is of broad interest to both methodologists and applied researchers who can learn from existing experience rather than starting from scratch. While interdisciplinary venues and HCI regularly feature such work [34, 24, 20, 29], there is a dearth of similar studies that get deep into the weeds about the nuts-and-bolts of different methods. We have had valuable off-the-record conversations with domain practitioners highlighting face (in-)validity of different methods and detailed choices of feature selection. Though this matters a lot in practice, we find little discussion in the research literature.

A distinct contribution occurs when the ML model itself is of primary interest to the application domain (e.g., a validated clinical prediction model). In this case, there may be a scientific contribution to the application domain, even if there is not for the field of machine learning more broadly. We believe that such papers are best reviewed by venues in the relevant application area, because experts in such domains are best positioned to evaluate the importance and utility of the contribution to the application domain. Machine learning venues, however, should welcome papers that use applied projects to draw scientific conclusions about the usage and impact of machine learning itself. As we argue below, machine learning as a field should also welcome researchers who build a portfolio of work spanning both application-focused contributions and papers of more general interest to ML.

## 3   Non-deployment contributions: methodology's paths to impact

How can general improvements in methodology have beneficial social impacts, even if a project is not deployed on-the-ground? Methodological work can be strong on different kinds of contributions that can sharpen and amplify social impact. Broader discussions about methodological impact and research priorities are standard in AI/ML; we do not rehash them here. Here, we focus on spelling

out some finer-grained criteria of how non-deployed methodological work can still advance machine learning for social impact, beyond standard justifications around solving novel problems and improved performance on benchmark data.

**Changing people's minds about estimation strategies and evaluations**   Formal analysis and methodological work can be the most impactful when it changes applied researchers' and data scientists' minds about how something should be done. While some theoretical analysis can live a full life in theory alone, and some methodological improvements can inspire future methodological innovations, methodology in the social impact domain only comes alive through use by others. The converse is also true - methodology that aims for challenges that social impact partners face, but does not manage to change applied researchers' minds about how to conduct analyses, achieves less impact. It therefore becomes important to understand and have empathy for the wide variety of disciplinary backgrounds from which data analysts at applied partners come from. This can include domain-centric training in the social, health, political, or other sciences, of which quantitative methods may be a small portion. Work whose key ideas and outputs can be translated to these domains reaches wider audiences and has more impact.

Advances in tooling can improve accessibility for complex methods (e.g., pretrained models for medical imaging). Nonetheless, it's easier to convince users to overcome learning curves when it opens new capabilities (e.g. pre-trained models for multimodal data), vs. when it competes with pre-existing methods that are seen as tried-and-true by applied communities (such as deep learning for tabular data vs. simpler methods). In the latter case, it's up to authors to make a convincing case for potential gains. This can include careful discussion about how inductive bias from new methodology can be relevant to a particular domain; or better yet, illustrating with re-analysis of prior empirical studies how new methods can shed new insights.

**Designing for maintenance**   Maintenance is systematically undervalued relative to innovation [30]. It's particularly important for methods designed for AI and social impact to be easy to maintain. Partner organizations in domains like public health, social services, etc. rarely have teams of ML PhD graduates who can maintain complex pipelines, for example. The simpler the tool is, the more likely it is to be adopted. [19] evaluates pilot studies of behavioral science-informed communication changes with public sector partners, and they find that the projects that are deployed beyond an initial pilot are those that leverage existing projects and infrastructure, rather than requiring additional resources to develop further. Who will maintain the system after students and PIs on an AI/ML project move on?

Although algorithmic decision support is widespread, depending on the domain, it often takes on a simple form. Consider nomograms - calculators for logistic regression coefficients that can be manually calculated by a physician at a patient's bedside [11]. Similarly simple tools like scorecards and categorical prioritization [27] are widely used in practice. Machine learning that restricts classification tools to take on this functional form can admit rich methodological work while aligning in form with what is used in practice, increasing the likelihood of deployment and adoption.

**The case for single-variable benchmarks**   Researchers should receive appropriate "credit", in terms of the assessed strength of contributions, for carefully establishing the difficulty or potential ease of a problem. Researchers should also consider exceedingly simple benchmarks, such as single-variable predictions, which recent works have shown have comparable performance as more complex ML approaches [35, 40, 38]. These benchmarks measure the marginal improvement of more complex machine learning relative to the simplest predictions possible, which can help differently resourced organizations leverage prediction. The specific model they use may depend on how much "total cost of ownership" they can afford to upkeep. Further, assessing the predictive performance of single-variable benchmarks supports transportability of findings. Often ML for social impact papers leverage data from a single organization, whose data columns and schema may or may not line up with data from other organizations. But reporting single-variable predictive performance can be readily replicated or compared for other organizations.

## 4   Recognizing more diverse contributions benefits everyone

**A healthy ML for social impact ecosystem cannot be sustained by only rewarding projects that achieve both deployment and methodological innovation**   Above, we have detailed many kinds

of research contributions, some more applied and some more methodological, some that should be publishable in machine learning and venues and some better suited to domain-specific journals. We believe that it is important for machine learning as a field to value papers of all of these kinds (e.g., in hiring, promotion, and tenure processes). In order for machine learning researchers to produce socially impactful work, they must be able to engage deeply and genuinely with partners. Such engagement requires starting projects without the certainty that they will produce a sufficiently novel algorithmic innovation. Some projects will end up introducing such technical innovations, some will contribute to the scientific study of ML's usage, and some will contribute purely to the application domain. The field as a whole benefits when researchers can build domain expertise and partnerships, knowing that contributions in any of these categories will be valued in the end.

Moreover, an implicit requirement to include technical innovation in every project distorts incentives: researcher may steer collaborations toward problems likely to yield methodological novelty even when those methods are not the partner's highest-value priority. Such incentives are a disservice to partners who invest limited time and resources in academic collaborations. They also decrease the likelihood that non-academics will see collaborations with machine learning researchers as worthwhile. This doesn't prevent the possibility of encountering unsolved technical problems during problem selection: machine learning researchers will of course be able to provide the most value when there are in fact problems which require their particular expertise. Rather, we believe that both the field and partner organizations are best served when researchers are rewarded for building a portfolio of different kinds of contributions, ranging from more methodological to more applied, without having to force particular projects towards a predefined path. Staking too much on individual projects making it to deployment can also have adverse consequences for researchers. For example, in some sensitive domains, deployment is subject to the idiosyncratic capacities and priorities of a few specific entities and executive leaders. Some degree of luck is involved, in addition to skill. Overall, although deployment is a useful ideal for AI work for social impact, adopting it as a metric to evaluate research can be unhealthy for the research ecosystem and have knock-on adverse effects.

As an illustration of these points, we argue that recognizing some of the finer contributions we discussed earlier requires ecosystem-level changes in practice.

**Normalize exceedingly simple, e.g. single-variable benchmarks.** For example, we argue that comparing against domain-driven single-variable baselines is currently disincentivized. Although reviewers are used to requesting comparison to strong baselines, these are usually expected to be other ML papers, rather than domain-driven justifications of predictive variables. Social impact application areas have high "Bayes error"; i.e. low signal-to-noise regimes due to individual variations. Strong performance of single-variable or simple benchmarks should neither be surprising nor diminish the value of ML-focused innovations. When simple baselines work well, this can be great for the ecosystem of ML for social impact, by minimizing barriers to deployment. But, this requires a shift in expectations and practices on all fronts: reviewers, authors, and publication venues.

**Finer-grained criteria articulate different dimensions of "novelty" and "significance" achieved by practice-oriented ML for social impact.** Many AI/ML venues already emphasize significance and novelty. Our analysis has surfaced important finer-grained criteria that can speak to the significance or novelty of non-deployed/methodologically innovative or deeply embedded in practice/non-novel ML for social impact work, even if it is not the Platonic ideal of deployed and methodologically innovative work. For example, the 2024 AAAI AISI track introduces a criteria on the "facilitation of follow-up work", but its finer gradations focus on ML reproducibility. We also suggest that follow-up work is facilitated when methods are themselves simple, easy to maintain, and includes potentially transferable information on the performance of simple benchmarks for a given domain.

We have highlighted how recognizing some of the finer-grained contributions we delineated in the prior sections could require ecosystem-level shifts on the part of authors, reviewers, and publication venues. For example, while review guidelines have expanded for major AI/ML conferences to include examples of contributions for more typical AI/ML papers [8], venues might also display examples of the AISI-specific contributions we discussed earlier.

# 5 Evaluating deployments of machine learning models

A final path to impact is the most traditional one: deploying a machine learning system in the real world. In short, our position is that papers reporting deployments or field experiments should adopt

higher standards of clarity and rigor. Valuing deployment alone, without rigorous assessments of impact, introduces harmful incentives to "toss ML models over the fence" into deployment and can suppress key post-deployment challenges. Improving the evaluative process is particularly important for field experiments because they directly impact participants. We argue that this impact implies a corresponding obligation for researchers: to treat the design and analysis process with seriousness, ensuring that participants' contributions to the experiment produce the maximum scientific benefit for society as a whole.

While there has been extensive attention to reproducibility issues in ML [13, 31, 36] that might undermine computational evaluations, evaluating ML models in deployment is an altogether different task, closer to causal inference; [28] makes a similar point for medical AI. The machine learning community does not need to reinvent this process from scratch: although algorithmically-driven interventions can sometimes raise complications in evaluation, most best practices can be drawn directly from other fields where experiments are a core component of the discipline [18, 10] (e.g., economics or medicine). Drawing on such best practices, we recommend principles for design and analysis across three deployment types.

## 5.1 Pilot tests

A pilot test is an initial, usually smaller-scale, deployment of a new intervention in order to assess its feasibility, acceptability to participants, and surface last-mile challenges and logistical difficulties ahead of a larger-scale evaluation. Pilot tests are essential because they reveal issues that are otherwise difficult to discover. The smaller scale, combined with close monitoring, also helps minimize harms to participants or partners if the system fails in an unexpected way. Importantly, pilot tests do not aim to provide a decisive evaluation of the impact of the system. The strengths of pilot tests lie in their smaller scale, adaptability, and avoidance of a rigid set of pre-specified outcomes to measure. All of these characteristics maximize the benefit of pilot tests for quick iteration but make the data typically inappropriate to draw rigorous conclusions regarding effectiveness. Authors should clearly identify when their work should be regarded as a pilot test. On the one hand, this is beneficial for authors because it relieves them of the burden of rigor associated with a full-scale evaluation (as detailed below). On the other hand, authors should in turn confine the main claims of the paper to quantities that pilot tests are well-suited to measure: whether the system proved feasible to deploy, what new obstacles or other learnings were uncovered, and how the results will shape future iteration. If the primary results of the paper concern measurements of impact, then the design and analysis must be sufficient to support such claims, which will typically require characteristics of experimental or quasi-experimental evaluations that we discuss below.

## 5.2 Randomized controlled trials.

In a randomized controlled trial (RCT), the researchers randomly assign participants to either be exposed to the machine learning system, or to one or more comparison conditions. The aim is to enable causal conclusions about whether exposure to the machine learning system improves participants' outcomes. We argue that RCTs of algorithms should adopt best practices for RCTs in other settings. These include:

**Preregistration:** before data collection begins, authors should publicly register a plan for the trial containing the protocol for how the trial will be run (e.g. sample size and eligibility criteria) and analysis strategy (e.g. what outcomes will be measured and how treatment effects will be estimated). We encourage authors and reviewers to consult established guidelines for information that should be included in preregistrations [22]. Preregistration has become a requirement to publish field experiments at many journals in order to ensure that trials have a clearly specified hypothesis and analysis plan; hunting for effects after the data is collected greatly increases the likelihood of false discoveries ("p-hacking") [33]. However, preregistration is not required at machine learning or AI conferences, or even covered in checklists like the NeurIPS Paper Checklist. We propose that machine learning publication venues should require that authors declare whether field experiments and analyses were preregistered or not. Non-preregistered analysis planned after data collection can still provide valuable findings but these should be explicitly distinguished in the text of the paper and reviewers should exercise additional skepticism regarding the robustness of these findings to alternative analytical choices.

**Power analysis:** as part of the process of developing a preanalysis plan and design, researchers should determine what effect size their proposed sample size will be able to detect with well-controlled type 1 and type 2 error. Underpowered trials yield noisy results and potentially waste effort from partner organizations and participants on trials that cannot detect plausible effect sizes.

**Outcome validity:** Researchers must justify outcome choices. In an ML context, outcome selection is often driven by what a digital platform already records; authors should explain how such proxies relate to welfare or consider collecting additional outcomes. For example, projects related to health sometimes measure how an intervention impacts engagement with a health messaging intervention because tracking how such engagement translates into downstream health outcomes is more difficult. Researchers should explicitly describe what evidence links these kinds of within-platform metrics to improvements in participants' welfare. Depending on the strength of this evidence, they should also weigh the costs and benefits of investing resources in gathering a wider range of outcome data beyond what is available by default. More broadly, recent work in machine learning has drawn attention to the social-scientific concept of validity in decisions like what outcome a model predicts [17, 26], and this scrutiny should be extended to the choice of outcomes for field experiments.

**Principled strategies for exploring heterogeneity:** often times, researchers want to know not just whether an intervention works on average, but who it works better or worse for. This is particularly relevant in the context of machine learning, which often aims to leverage such heterogeneity. Detecting heterogeneous effects is, however, quite data-intensive compared to measuring averages and is particularly prone to generating false positive results due to the number of comparisons made. Researchers should preregister strategies that they will use to detect heterogeneity, whether via explicit subgroup analyses, or a specific algorithmic procedure.

So far, these principles are standard best practices for RCTs that we argue should be adopted by authors and enforced by reviewers and publication venues. Further, RCTs specifically of ML or other algorithmic systems can require additional complexity, two of which we specifically note below.

First, the appropriate unit of randomization may differ depending on how the system is used. If treatment is assigned to individuals (e.g., the algorithm provides an individual diagnosis or recommendation to each individual), then randomization can be at the individual level. On the other hand, algorithms sometimes operate at a group level: they may be deployed at the level of a service center, hospital, school, etc, without the ability to cleanly limit the exposure to the algorithm to specific individuals within the group. In this case, multiple different groups must be randomized to either receive the algorithm or not.

Second, algorithms often function as a resource allocation mechanism for another intervention, instead of serving directly as an intervention themselves. This setup presents the researcher with a dilemma. A resource allocation system operates at the level of the population. Accordingly, as discussed above, in principle the RCT should also randomize at a group level, assigning some populations to have resources allocated algorithmically and others in an alternative fashion. However, conducting this ideal trial may be quite difficult in practice, particularly when the implementing partner consists only of a single "group". Under additional assumptions, recent strategies use data from a single population to estimate treatment effects on those the algorithm selects (i.e., whether it selects higher-benefit individuals) [15, 25, 43].

More broadly, this point illustrates how evaluations of algorithmic systems can surface new challenges in the design and analysis of randomized trials. See also, for example, questions about how the medical regulatory process should treat machine learning [21, 16]. Developing improved strategies to empirically test and evaluate interventions that are mediated by algorithms is itself a promising area for future research.

### 5.3 Non-randomized deployments

In some cases, conducting a randomized trial is infeasible, perhaps because randomization is seen as inappropriate or undesirable to a partner organization. In such cases, researchers often report before–after comparisons of outcomes pre- and post-launch. We argue that these comparisons should be understood as an instance of an observational design known as an *event study* [23, 32]. The primary threat to internal validity is that the before and after time periods are different in some way unrelated to the introduction of the algorithm, resulting in a difference in outcomes that is not explained by the

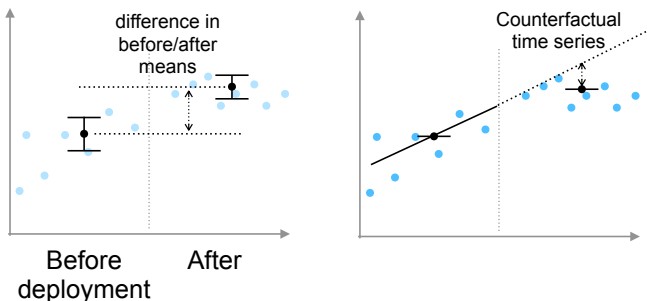

Figure 1: Non-randomized deployments and event -studies

introduction of the machine learning systems. A large and rapidly growing subfield of econometrics research studies different variants of event study designs, whose complexity depends on the extent of information available for potential control groups; no control (interrupted time series), one control unit (differences-in-differences) vs. many comparable control units (synthetic control).

For now, we assume only access to data from the organization deploying the AI/ML (i.e. no control), and illustrate how naive before-and-after comparisons can be confounded by systematic changes (over time or in covariates), and how basic ideas like interrupted-time-series improve upon this by introducing the idea of a *counterfactual* – What *would have happened* in the *absence* of deployment? Figure 1 illustrates this. The y-axis reflects repeated measures of the outcome variable (which presumably deployment improves), with the x-axis indicating the passage of time. A naive non-randomized deployment compares outcomes prior to the deployment at time $t_0$ with outcomes after the deployment. On the left is a typical simple before-after comparison of outcomes: estimating causal impacts of deployment by comparing differences in average outcomes before and after deployment. The next subplot illustrates how temporal structure can undermine such simple before-after comparisons. Taking a look at the time series of outcomes prior to deployment, outcomes had been improving prior to deployment. The simplest event study, called interrupted time series, compares post-deployment outcomes to a *counterfactual time series* based on predicting *what would have happened in the time after $t_0$, if the system had not been in fact deployed.* In this hypothetical example, the organization may have been on an upward trend of improving outcomes (or there is seasonality). Compared to the extrapolation of pre-deployment outcome *trends*, average post-deployment outcomes actually indicate a net *negative* causal effect relative to *the counterfactual time series*.

Even in the absence of an explicit control group in a non-randomized deployment, one can and should reason more carefully about counterfactual outcomes. In this case, it requires predictive modeling based on pre-deployment data, but the task of generalization and prediction is familiar to machine learning. More advanced event study designs require additional information, such as *differences-in-differences* with an additional control unit and additional assumption of *parallel trends*, that the difference in outcomes between treatment and control groups is constant in time. Minimally, machine learning evaluations employing an event study should report quantities like pre-deployment outcome trends and before-after changes in the composition of the population in order for readers to judge the importance of factors besides the deployment of the algorithm itself.

## 6 Alternative views

We acknowledge potential alternative views or counterarguments, responding to them in turn.

1. Machine learning for social impact should be hosted only in domain-specific academic departments and publication venues.

   This would be tragic for the field of machine learning, as designing methodology to solve problems for application areas can be a rich design constraint to advance machine learning fundamentals.

2. Machine learning for social impact should be reviewed like any other paper; otherwise readers might downweight and devalue AISI papers.

   We view different tracks for ML for social impact at major conferences as a positive development, recognizing that different kinds of contributions require different criteria and reviewing processes.

3. Major conferences have required impact statements; these suffice for AISI.

   Impact statements help all papers reflect upon their paths to impact and are particularly helpful in recognizing otherwise unseen impacts to methods-focused work. But projects specifically targeting social impact should reflect different design evaluation choices in methodology and findings throughout the whole project, which we have articulated here.

4. Early-career researchers should be strategic and prioritize projects that do lead to widespread recognition, e.g. those that are deployed and methodologically innovative.

   Different research teams have different interests and comparative advantages. But much work in ML for social impact advertises its significance based on high-stakes consequences. We argue this imposes moral obligations to responsibly orient work towards improving impacts in the domain for affected individuals, even if at times this may not align with career currency. At the same time, we believe it's necessary to cultivate ecosystems that *do* align incentives for deeper partnerships and domain-level social impacts, to the extent possible. Else, AISI risks becoming a PR project for the field of computing alone.

5. The field should instead progress via "machine learning + X" where X is the application area, like ML+Health, ML+Science, ML+Ecology, etc ...

   While such ML+X venues are great for the community, there are separate benefits to including machine learning for social impact works at general venues. Fracturing the work along lines of application domains prevents generalizing lessons from specific domains to more general challenges potentially faced in many domains, which may not be obvious from the point of view of theory alone.

## 7  Conclusion and recommendations

The considerations discussed in the previous four sections entail recommendations for participants across the research ecosystem. Authors should articulate mechanisms of impact underlying a given project, which may be any of the different kinds of contributions distinguished above. Reviewers and publication venues should explicitly adopt an expansive notion of contribution: AISI projects may be stronger on some criteria than others. Even if some of these criteria may currently be *implicit* values in the AISI research community, we strongly believe that discussing them *explicitly* is necessary to foster a sustainable research ecosystem. Making this process explicit also helps reviewers demand high-quality evidence for the kind of contribution that authors claimed. Improving the evaluative process is particularly important for field experiments, which directly impact participants, and we recommend that reviewers and publication venues adopt additional guidelines specifically for such deployments. Broadly, we argue that individual researchers face a coordination problem where many kinds of AISI contributions are underappreciated, distorting the way that research is presented and evaluated. Collective action by authors, reviewers, and publication venues can create a healthier ecosystem for the entire field.

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
