# OpenReview forum: "Fostering the Ecosystem of AI for Social Impact Requires Expanding and Strengthening Evaluation Standards"
_NeurIPS.cc/2025/Position_Paper_Track — NeurIPS 2025 Position Paper Track_

### Official Review · Reviewer_p8qc · 2025-07-10

**Significance:** 4
**Presentation:** 3
**Rating:** 8
**Confidence:** 4

**Summary:**

This paper critiques the dominant research incentives in AI for social impact (AISI), which prioritize projects combining the deployment of novel methods. The authors argue this approach neglects valuable contributions, including:

1. Partner organizations often benefit from researchers applying existing ML tools without innovation, fostering domain expertise and sustainable solutions.

2. Novel techniques may lack deployment but still advance AISI by addressing domain-specific challenges or redefining evaluation standards.

3. Current practices often lack causal rigor, with weak validation of claimed impacts.

The paper proposes three reforms:

1. Expand impact definitions to include non-deployment contributions (e.g., domain-specific ML applications).

2. Strengthen evaluation standards for deployed systems, adopting principles from randomized controlled trials (RCTs) and quasi-experimental designs.

3. Normalize simple benchmarks (e.g., single-variable predictions) to compare against ML innovations, enabling resource-constrained organizations to adopt solutions.

**Strengths:**

1. Bridges AI, social sciences, and policy to articulate AISI’s unique challenges.

2. Proposes concrete reforms (e.g., preregistration for field trials, single-variable benchmarks).

3. Addresses resource constraints and partner organization needs, emphasizing maintainable solutions.

4. Highlights how current incentives distort research priorities, benefiting neither partners nor the field

**Weaknesses:**

1. Lacks case studies demonstrating the effectiveness of proposed evaluation frameworks.

2. Some critiques of benchmarking may not account for recent advances in contextual evaluation.

3. Limited discussion of how to standardize "simple benchmarks" across domains or enforce preregistration in conferences.

4. Does not fully address how to balance domain-specific vs. general ML tracks without fragmenting the field

**Questions:**

1. How would you operationalize "facilitation of follow-up work" as a review criterion for AISI projects?

2. What mechanisms could ensure preregistration adoption without stifling exploratory fieldwork in resource-limited settings?

**Alternative Position:**

Yes, and alternative positions are well-considered and named but not addressed

**Author Identification:**

No.

**Context:**

4

**Details Of Ethics Concerns:**

This theoretical position paper focuses on evaluation frameworks and research incentives in AI for social impact. It does not involve human subjects, data collection, or system deployment. The work is conceptual, addressing structural challenges in the academic ecosystem rather than ethical dilemmas in AI applications

**Discussion:**

4

**Ethics:**

["NO or VERY MINOR ethics concerns only"]

**Position:**

Yes, the paper argues for or against a position related to machine learning.

**Support:**

4

**Thoroughness:**

4

---

### Official Review · Reviewer_qBf7 · 2025-08-01

**Significance:** 4
**Presentation:** 3
**Rating:** 7
**Confidence:** 3

**Summary:**

This position paper critiques the current norms in machine learning for social impact (AISI), particularly the overemphasis on projects that combine both deployment and methodological novelty. The authors argue that such criteria create narrow incentives and fail to recognize valuable contributions that are either applied (e.g., supporting partners in using standard ML tools) or methodological (e.g., proposing approaches that haven't yet been deployed). The paper calls for the AISI community, including authors, reviewers, and venues, to broaden what is considered impactful research. It proposes three reforms: (1) recognizing practice-driven work without novel methods, (2) valuing non-deployed methodological work that has plausible real-world relevance, and (3) requiring more rigorous evaluations of deployed systems. Drawing on examples and review criteria from existing AISI tracks, the paper advocates for a more inclusive, sustainable research ecosystem that aligns scientific and social value.

**Strengths:**

## Strengths of the paper

The paper highlights a critical and timely issue in the AI/ML for Social Impact (AISI) community, the overemphasis on deployed, methodologically novel work in current review and publication processes. This reflection on the field's incentives is highly relevant as AISI matures and expands. The authors present a well-structured and multi-faceted argument, encouraging the community to value diverse contributions, such as applied collaborations without new methods and methodological research without immediate deployment. The call for more rigorous evaluation of real-world deployments, including clear distinctions between pilot tests, RCTs, and non-randomized studies, is especially valuable and under-discussed. The paper also thoughtfully engages with alternative viewpoints, making the discussion more balanced and comprehensive. Overall, the paper raises awareness about structural imbalances in research recognition and offers a foundation for improving how impactful work is identified and rewarded.

**Weaknesses:**

## Weaknesses

While the paper raises several valid and timely concerns, the depth and specificity of its proposed solutions fall short in several sections.

**Section 2 (Non-method contributions):** The concern is valid, but the discussion lacks clarity on what distinguishes meaningful ML contributions from pure implementation. Examples or review guidelines would have helped.

**Section 3 (Non-deployment contributions):** The argument is important, but remains abstract. Concrete cases where methodological work had downstream impact without deployment would make the point stronger.

**Section 5 (Evaluating deployments):** This is a rich section but quite dense. Concepts like RCTs and causal inference are discussed assuming prior knowledge. It would benefit from simplified language or illustrative templates to make it more accessible to the broader ML community.

**Overall:** The paper identifies problems well but offers few concrete, actionable solutions. A checklist or decision framework for reviewers and authors could significantly improve its practical utility.

**Questions:**

## Questions

The paper is a great read and highlights key challenges in current AISI tracks. It offers a thoughtful, high-level analysis of how the community might address them. I have a few questions to better understand and contextualize some of the proposals:

**Section 2 (Non-method contributions):**
How can reviewers distinguish between meaningful ML research that supports partners and routine engineering work? Are there specific criteria or signals you recommend?

**Section 3 (Non-deployment contributions):**
While non-deployed work can be impactful, should some domains (e.g., healthcare, safety-critical systems) emphasize deployment more? How should review criteria reflect this?

**Section 5 (Evaluating deployments):**
This section is insightful but dense. How should ML researchers unfamiliar with causal inference approach RCTs or time-series evaluations? Are there practical resources or templates you recommend?

**Section 7 (Recommendations):**
You propose that authors state a “theory of change.” Could you give an example of what this looks like in practice, and how it might be evaluated in peer review?

**Alternative Position:**

Yes, and alternative positions are well-considered and addressed by the argument

**Author Identification:**

No.

**Context:**

3

**Discussion:**

4

**Ethics:**

["NO or VERY MINOR ethics concerns only"]

**Position:**

Yes, the paper argues for or against a position related to machine learning.

**Support:**

3

**Thoroughness:**

3

---

### Official Review · Reviewer_ybiQ · 2025-08-03

**Significance:** 4
**Presentation:** 4
**Rating:** 9
**Confidence:** 5

**Summary:**

This position paper highlights that the AI for Social Impact (AISI) research community's current evaluation criteria are too fixated on awarding projects that are both methodologically new and successfully implemented. The authors argue that this gap creates unhealthy incentves and undervalues other vital forms of contribution that are often overlooked.
The primary focus of the paper advocates for redefining the evaluation criteria framework , which includes: 1) changing the evaluation criteria to a broader conception of impact and rigorously applying methods that involve non-deployed method-work that is constrained by the real-world, and 2) applying stricter evaluation criteria to deployed systems, building from empirical fields like preregistered randomized controlled trials (RCTs) and non-randomized event studies.
The main contribution of the paper is this framework designed for authors, reviewers, and venues to cultivate a healthier, more sustainable and more genuinely impactful research ecosystem for AISI.

**Strengths:**

This paper is remarkable for clearly defining and defending a substantial position: that the AI for Social Impact (AISI) community needs to reconsider the scope of its contribution and simultaneously refine its impact assessment criteria of its systems.

The authors provide a thorough explanation on the reasoning, which is grounded in a wide variety of social and community practices. By citing community practices, the authors build a compelling narrative that is crucial for the community to embrace more advanced evaluation criteria. Most importantly, the authors provide explanation of advanced evaluation techniques from other disciplines (e.g. preregistration, and event studies) which demonstrates that the recommendations are based on evidence.

This subject is very important to the NeurIPS community. As AISI grows into a more defined track, the community needs to address a defining a measuring impact and contribution to the community. This paper offers a well-rounded, constructive and actionable framework to enhance the trustworthiness and impact of the work presented at NeurIPS and other venues. It is stands out position paper for its model of clarity and rigor.

**Weaknesses:**

The authors of the document should explain how disregarding funding agencies as a target audience is a missed opportunity for advancing the discussion of research incentives as a whole. Addressing factors like promotion structures within universities, which also serve as major motivators of research, is a largely unexplored area that, in combination with the proposed evaluation framework, could greatly enhance the argument. In a bid to provide more value for the community, the authors could issue a checklist or a collection of central guiding questions that would distill the reviewer's work to the essence of the proposed framework in the appendix.

The authors could adopt a more radical stance that high-ranking ML conferences should only accept papers featuring rigorously evaluated, large-scale deployment of Artificial Intelligence Systems, relegating all other forms of contribution to specialized workshops or domain-specific journals. This would serve as a counterargument to what authors call for an “expansive ecosystem” dominating the ML community, which would shift the bulk of prestige and attention to the other side of the field.

**Questions:**

I appreciate you presenting this relevant and insightful paper. I would like to raise a couple of queries regarding the practical aspects of framing your vision:

Your critique regarding the lack of robust evaluation on deployments utilizing causal inference approaches is indeed a strong one. However, this contains a single point of failure: the shortage of ML evaluators. How do you recommend conference organizers deal with this? Should we assign unique sub-reviewers to the evaluation parts, or is the objective to raise the education level of the entire reviewer pool?

Along the same line of thinking, what steps can we as a collective take to enforce that the threshold for 'non-methodological' contributions does not devolve into a free for all application case study? Providing a definitive answer to this question would further support your position that these contributions ought to be recognized and discussed in prestigious ML conferences.

**Alternative Position:**

Yes, and alternative positions are well-considered and addressed by the argument

**Author Identification:**

No.

**Context:**

4

**Discussion:**

4

**Ethics:**

["NO or VERY MINOR ethics concerns only"]

**Position:**

Yes, the paper argues for or against a position related to machine learning.

**Support:**

4

**Thoroughness:**

5

---

### Note · Authors · 2025-08-21

**1-11 Submit Again:**

Probably yes

**1-1 Submission Process:**

5

**1-4 Interest:**

["Panel discussions with other position paper authors", "Workshops for developing position papers", "Mentorship programs for early-career researchers"]

**1-5 Thoughtful:**

8

**1-6 Supportive:**

7

**1-7 Technical Aspects Versus Position:**

8

**1-8 Gate Keeping:**

8

**1-9 Camera Ready Changes:**

See our responses to Section 3 below: we have included responses to Reviewer Questions, which help us clarify additional context we can provide, or additional clarifications we can make. We will add these to the camera-ready version.

**3-1 Review Response1:**

ybiQ

**3-2 Reaction To Review1:**

We were overall appreciative and satisfied with the reviews, which we thought were thoughtful and fair. We'd like to use this opportunity to respond to questions posed in the review, and provide clarifications that we will add in the camera-ready, in order to further develop and operationalize our positoin.

ybiQ asks: "Your critique regarding the lack of robust evaluation on deployments utilizing causal inference approaches is indeed a strong one. ... How do you recommend conference organizers deal with this? Should we assign unique sub-reviewers to the evaluation parts, or is the objective to raise the education level of the entire reviewer pool?"

It’s impractical to assign unique sub-reviewers to evaluation – we suggest to raise the education level of the entire reviewer pool, just as in ML best evaluation practices on cross-validation and standard errors are standard. AI for social impact should be accompanied with general education as to best practices to measuring and establishing magnitude of impacts.

"Along the same line of thinking, what steps can we as a collective take to enforce that the threshold for 'non-methodological' contributions does not devolve into a free for all application case study? Providing a definitive answer to this question would further support your position that these contributions ought to be recognized and discussed in prestigious ML conferences."

This is an inherent challenge to interdisciplinary work. One way of supporting this is to establish norms of inviting external reviewers or subreviewers who may be domain experts. Domain experts are well-equipped with the taste to understand whether findings from non-methodological contributions are important, timely, relevant, or otherwise interesting to the applied field. Area chairs at the boundaries of “ML and X” might be able to reach out to domain experts with targeted inquiries as to the relevance or significance of non-methodological contributions to the applied field.

**3-3 Review Response2:**

​​qBf7

**3-4 Reaction To Review2:**

qBf7 asks: "How can reviewers distinguish between meaningful ML research that supports partners and routine engineering work? Are there specific criteria or signals you recommend?"

Our proposal is that “meaningful” research still needs to make a scientific contribution; the community should just embrace a wider set of possible topic areas for that contribution. The examples in section 2 underpin a criteria we propose for ML publication venues: significance can be established by arguing that a paper improves our understanding of how machine learning is used by individuals or organizations, what the impacts of that usage are, and which design choices in machine learning shape that impact. Authors should contextualize how their empirical work with a partner advances knowledge compared to the existing literature. One strategy we suggest is to openly discuss problem formulation, including potentially discarded problem formulations.

Section 3 (Non-deployment contributions):
"While non-deployed work can be impactful, should some domains (e.g., healthcare, safety-critical systems) emphasize deployment more? How should review criteria reflect this?"
Your question as to different domains points to the differing barriers for deployment in different fields. Expectations re: deployment should be different for different fields. This can be part of appropriate positioning in related work.

Section 5 (Evaluating deployments):"How should ML researchers unfamiliar with causal inference approach RCTs or time-series evaluations?"
We recommend standard textbooks on causal inference with particular attention to program evaluation methods (e.g. Angrist and Pischke’s “Mostly harmless econometrics”, since they are most often used to assess the effects of causes (like a new algorithm or deployment). Also JPAL’s Resources page for randomized evaluationsand the Causal Inference Mixtape for observational studies (https://mixtape.scunning.com/). Will add.

**3-5 Review Response3:**

p8qc

**3-6 Reaction To Review3:**

p8qc asks: "How would you operationalize "facilitation of follow-up work" as a review criterion for AISI projects?

We think this can be operationalized via 1) reproducibility guidelines – how easy is the code to use? and 2) considering ease of deployment and maintenance as contributing to the potential significance of the work. This rewards simplicity, even if something is not technically demand.

p8qc asks: "What mechanisms could ensure preregistration adoption without stifling exploratory fieldwork in resource-limited settings?"

The “pilot” category of projects is a good category for exploratory fieldwork.

---

### Meta-Review · Area_Chair_i3tU · 2025-09-18

**Rating:** 8
**Confidence:** 4

**Strengths:**

Reviewers had a high opinion of the paper, agreeing that it made a clear and cogent argument that the AI for Social Impact (AISI) community should relax the requirement that good work must always be both deployed and methodologically innovative. Its argument is well supported with evidence and the paper makes clear what the community would have to do in order to improve on the aspects it highlights.

**Weaknesses:**

The authors have promised to address some of the points raised by reviewers, leading one reviewer to increase their score. Reviewers would like to see additional concrete examples and case studies to illustrate abstract points. Reviewers also would like to see some potential counter-arguments rebutted, including the concern that the proposal might further fragment the NeurIPS community.

**Questions:**

Multiple reviewers raised questions about the evaluation of non-methodological contributions. How could the evaluations proposed be more specific and actionable?

**Ethics:**

None raised

**Thoroughness:**

4

---

### Decision · Program_Chairs · 2025-09-26

Accept